# And yet Again: Having Breakfast Is Positively Associated with Lower BMI and Healthier General Eating Behavior in Schoolchildren

**DOI:** 10.3390/nu13041351

**Published:** 2021-04-18

**Authors:** Peggy Ober, Carolin Sobek, Nancy Stein, Ulrike Spielau, Sarah Abel, Wieland Kiess, Christof Meigen, Tanja Poulain, Ulrike Igel, Tobias Lipek, Mandy Vogel

**Affiliations:** 1LIFE Child, Hospital for Children and Adolescents, Medical Faculty, Leipzig University, Ph-Rosenthal-Str. 27, 04103 Leipzig, Germany; Peggy.Ober@medizin.uni-leipzig.de (P.O.); Sarah.Abel@medizin.uni-leipzig.de (S.A.); Wieland.Kiess@medizin.uni-leipzig.de (W.K.); Christof.Meigen@medizin.uni-leipzig.de (C.M.); Tanja.Poulain@medizin.uni-leipzig.de (T.P.); ulrike.igel@fh-erfurt.de (U.I.); Tobias.Lipek@medizin.uni-leipzig.de (T.L.); Mandy.Vogel@medizin.uni-leipzig.de (M.V.); 2Center for Pediatric Research (CPL), Hospital for Children and Adolescents, Medical Faculty, Leipzig University, Liebigstr. 20a, 04103 Leipzig, Germany; Ulrike.Spielau@medizin.uni-leipzig.de; 3Integrated Research and Treatment Center Adiposity Diseases, Medical Faculty, Leipzig University, Ph.-Rosenthal-Str. 27, 04103 Leipzig, Germany; nancy_stein94@web.de; 4Department of Social Work, University of Applied Science, Altonaer Str. 25, 99085 Erfurt, Germany

**Keywords:** overweight, obesity, breakfast, breakfast skipping, lunch, meal frequency, children, school

## Abstract

Given the high prevalence of childhood overweight, school-based programs aiming at nutritional behavior may be a good starting point for community-based interventions. Therefore, we investigated associations between school-related meal patterns and weight status in 1215 schoolchildren. Anthropometry was performed on-site in schools. Children reported their meal habits, and parents provided family-related information via questionnaires. Associations between nutritional behavior and weight status were estimated using hierarchical linear and logistic regression. Analyses were adjusted for age, socio–economic status, school type, migration background, and parental weight status. Having breakfast was associated with a lower BMI-SDS (*β*_adj_ = −0.51, *p* = 0.004) and a lower risk of being overweight (OR_ad_j = 0.30, *p* = 0.009), while having two breakfasts resulting in stronger associations (BMI-SDS: *β*_adj_ = −0.66, *p* < 0.001; risk of overweight: OR_adj_ = 0.22, *p* = 0.001). Likewise, children who regularly skipped breakfast on school days showed stronger associations (BMI-SDS: *β* = 0.49, *p* < 0.001; risk of overweight: OR = 3.29, *p* < 0.001) than children who skipped breakfast only occasionally (BMI-SDS: *β* = 0.43, *p* < 0.001; risk of overweight: OR = 2.72, *p* = 0.032). The associations persisted after controlling for parental SES and weight status. Therefore, our data confirm the school setting as a suitable starting point for community-based interventions and may underline the necessity of national programs providing free breakfast and lunch to children.

## 1. Introduction

Childhood obesity is a major public health challenge for societies worldwide. The prevalence stagnated at high levels after significantly increasing since the 1980s [1,2,3]. In Germany, 15.4% of children and adolescents are overweight, and 5.9% are obese [4]. As obese children are very likely to remain obese later in life [5], often afflicted with a multitude of sequelae, the persistent high prevalence of childhood obesity poses an enormous economic burden on societies [6].

Overweight is strongly linked to social inequalities. In western societies, the proportion of overweight and obesity is highest among children and adolescents of low socio–economic status (SES) [4,7,8]. Hence, the SES is one part of a multifactorial obesogenic environment to which children are exposed [9,10]. In addition, high intakes of energy-providing food and beverages, high media use, and parental overweight are relevant risk factors for childhood obesity [11,12]. These risk factors are linked to the home environment and are therefore very difficult to address with preventive interventions, especially in low SES families [13].

Similarly, whether and how meals are eaten depends on family habits and parental behavior [14,15]. However, during a school week, children and adolescents spend most of their waking time at school. Therefore, breakfast and lunch behavior is partly dependent on the schedule and food availability within the school environment. Interventions in schools are a cost-effective opportunity to promote healthy behaviors and to create a healthy environment [16]. Thus, the school setting offers a particularly suitable starting point for interventions because of its unique potential to reach children of all social backgrounds, independently of their home environment.

To facilitate tailored interventions, this study aimed to examine the nutritional behavior of schoolchildren and its association with weight status.

## 2. Materials and Methods

### 2.1. Study Design and Sample

In the cross-sectional Leipzig school nutrition study, we collected data in public and private schools in Leipzig, Germany. The data were collected between May 2018 and May 2019. Schools were chosen from predefined areas within Leipzig based on socio–economic data and overweight prevalence as described by Igel et al. [17]. All schools within these areas (*n* = 42) were eligible. After the approval of the school administration, all classes in grade four (elementary schools) and grades 6–8 (secondary schools) were invited to participate. Informed consent was obtained from all parents of participating children (rates can be seen in Figure 1). Finally, 34 schools participated, and 1215 children and adolescents were recruited there. The schools did not receive any financial benefits. The children received a financial incentive of five euros. The study was approved by the Ethical Committee at the Medical Faculty of the Leipzig University (number 483/17-ek).

### 2.2. Setting

The survey took place in the schools. The study leader led the children through the entire survey. The parents were asked about family socio–demographic characteristics through a questionnaire completed at home.

### 2.3. Instruments

Body height was measured, without wearing shoes, with an accuracy of 0.1 cm, using a portable stadiometer (Seca, Hamburg, Germany). Body weight was measured with an accuracy of 0.1 kg, wearing underwear and one layer of top clothing, with a calibrated electronic scale (Kern, Balingen, Germany). Weight was corrected for the estimated weight of the clothing [18,19]. Subsequently, the body mass index (BMI) was calculated and transformed to standard deviation scores (SDS) using the references from Kromeyer-Hauschild [20], according to the German guidelines of the German Working Group of Obesity in Childhood and Adolescents [21]. SDS allowed the assessment of weight status in the context of age and sex by transforming a BMI value into its deviation from the age- and sex-dependent mean expectation, measured in multiples of a standard deviation. SDS are equivalent to Z-scores [22]. The weight groups were defined as normal weight (BMI-SDS <1.28) and overweight/obese (BMI-SDS ≥1.28) [20,21]. All procedures were conducted by a trained pediatric nurse following standard protocols.

Breakfast habits were assessed by asking children if they usually eat breakfast on school days. Response options were “No”/“Yes”, “at home before I go to school”/“Yes, at school—I buy something”/“Yes, at school—I bring something from home”. Multiple answers were allowed. Children who only checked the “No” option were defined as regular breakfast skippers, whereas children who checked “No” and at least one of the other options were defined as occasional breakfast skippers. The number of breakfasts was defined as “0” if the children were regular breakfast skippers and “2” if the children had one breakfast at home and one at school. Otherwise, the number of breakfasts was set to “1”. Lunchtime behavior was assessed by asking the children what they usually eat for lunch on school days. Response options were “I buy something on the way or outside school area”/“I eat what my parents give me”/“I eat what is offered in the school canteen”/“I eat nothing for lunch at school”. Multiple answers were allowed. Children who checked the “I eat nothing for lunch at school” option were defined as lunch skippers, whereas children who checked one of the remaining options were grouped as lunch consumers.

Information on the parents’ highest level of education (general and vocational), the current occupational position, and monthly equivalized disposable household income were combined to assess the SES as a modified Winkler Index [23]. The resulting score ranged between three and 21 and was classified into low (<8.5), moderate (8.5–15.4), and high (>15.4) according to the KiGGS study [24], a current representative study of German children and adolescents. Additionally, the migration background was asked for (“Yes” if at least one of the parents was born abroad). Self-reported weight and height of both parents were used to calculate parental BMI. According to the World Health Organization (WHO), subjects having a BMI ≥ 25 kg/m^2^ were classified as having overweight [25].

### 2.4. Statistical Analysis

Descriptive statistics were given as the mean and standard deviation for continuous variables and counts and percentages for categorical variables. Associations between breakfast skipping, having lunch, and overweight as dependent variables and age, SES score, SES group, school type, migration background, and parental weight status as independent variables were analyzed using hierarchical logistic regression models. Correspondingly, the hierarchical linear regression models were applied to assess the associations between BMI-SDS and the covariates. Furthermore, the association of overweight, BMI-SDS, and having lunch as the outcome and breakfast skipping and the number of breakfasts per day were examined. Finally, a hierarchical logistic regression analysis was performed with overweight/obesity as dependent and having no lunch as the independent variable. After assessing the bivariate associations, multivariate models were built based on the full model containing the covariates mentioned above using stepwise backward deletion. Models were adjusted for age and sex if necessary. To account for clustering effects, the school was added as a random effect. Due to the high collinearity leading to variance inflation, models containing the school type as a covariate were not adjusted for age. Additionally, due to a strong dependence between school type and the school itself, the models with school type as the only covariate were modeled without random effect. In addition, interactions between the covariates were tested. Study data were collected and managed using REDCap electronic data capture tools [26]. All analyses were conducted using R version 4.0 [27]. The level of significance was set to *α* = 0.05.

## 3. Results

### 3.1. Characteristics of the Study Sample

Thirty-four schools were recruited for the study (Figure 1). From these schools, 3107 children were eligible for participation. The participation rates varied considerably between the school types. In elementary schools, 43.6% (*n* = 690) of students participated. In secondary schools, attendance varied between 26.9% (*n* = 178) for lower secondary schools and 47.2% (*n* = 347) for upper secondary schools (usually leading to tertiary education). The parents´ questionnaire response rate was 88% (*n* = 607)/73.6% (*n* = 131)/86.2% (*n* = 299) for elementary/lower secondary/upper secondary schools. Finally, 1215 students were studied in school, of which 1037 parents´ questionnaires (85%) were returned. The mean age of the total sample was 11.32 years (SD = 1.35, range: 8.94–15.42), 617 were girls, and the BMI-SDS was *µ* = −0.04 (SD = 1.08, range: −3.58–2.99). The prevalence of overweight/obesity was 12.2% (*n* = 148). A more detailed description of the study population characteristics is shown in Table 1.

### 3.2. Breakfast and Lunch Behavior

More than half of the students (67.5%, *n* = 820) ate breakfast at home before school, and 76.5% (*n* = 860) reported buying something or bringing something from home for breakfast at school. Considering breakfast frequency, 47% (*n* = 569) of children had breakfast once a day and a further 47% (*n* = 569) of the children had breakfast twice a school day—once at home and once at school. Among the children studied, 8.1% (*n* = 98) often did not eat breakfast, neither at home nor at school. Among these, about a third (*n* = 26) were occasional breakfast skippers (had breakfast on some days), and 72 were regular breakfast skippers. Parameters associated with regular breakfast skipping are summarized in Table 2. Older children skipped breakfast more often than younger children (OR = 1.54, 95%CI: 1.27–1.87, *p* < 0.001). Girls skipped breakfast more often than boys (OR = 1.23, 95%CI: 1.23–1.24, *p* < 0.001), and the migration background was also related to higher skipping rates (OR_adj_ = 1.70, 95%CI: 1.69–1.71, *p* < 0.001). SES was inversely related to breakfast skipping, with lower skipping in children with higher SES (OR = 0.84, 95%CI: 0.77–0.93, *p* < 0.001). Importantly, this association persisted after adjustment (OR_adj_ = 0.85, 95%CI: 0.77–0.93, *p* < 0.001). We did not find a statistically significant association between parental overweight and breakfast skipping, maybe due to a high number of missing parental overweight status (*n* = 321).

Most of the students (83.4%, *n* = 1013) ate something for lunch. For nearly all of them, lunch was the second main meal of the school day (96.0%, *n* = 972). The probability that a child eats lunch during the school day increased with having breakfast. Children having one breakfast were more likely to have lunch compared to breakfast skippers (OR = 3.79, 95%CI: 2.17–6.62, *p* < 0.001). For children having two breakfasts, the association was even stronger with OR = 5.13 (95%CI: 2.89–9.10, *p* < 0.001). These associations persisted after adjustment (one breakfast: OR_adj_ = 3.89, 95%CI: 1.90–7.96, *p* < 0.001; two breakfasts: OR_adj_ = 5.25, 95%CI: 2.52–10.90, *p* < 0.001).

### 3.3. Association of Breakfast and Lunch Behavior with Overweight

Children having one breakfast on school days (whether at home or at school) had on average a 0.5 lower BMI-SDS than children who had not (*β*_adj_ = −0.51, 95%CI: −0.86–−0.15, *p* = 0.004) and were significantly less likely to be overweight (OR_adj_ = 0.30, 95%CI: 0.12–0.76, *p* = 0.009). The associations were moderately stronger when children had two breakfasts (BMI-SDS: *β*_adj_ = −0.66, 95%CI: −1.02–−0.31, *p* < 0.001; risk of overweight: OR_adj_ = 0.22, 95%CI: 0.09–0.56, *p* = 0.001). In addition, children who regularly skipped breakfast on school days showed stronger associations (BMI-SDS: *β* = 0.49, 95%CI: 0.23–0.75, *p* < 0.001; risk of overweight: OR = 3.29, 95%CI: 1.86–5.82, *p* < 0.001) than children who skipped breakfast only occasionally (BMI-SDS: *β* = 0.43, 95%CI: 0.20–0.66, *p* < 0.001; risk of overweight: OR = 2.72, 95%CI: −1.07–−6.93, *p* = 0.032). Moreover, having lunch was associated with lower BMI-SDS and a lower risk of being overweight, but the association lost statistical significance after adjustment. Unadjusted and adjusted associations are shown in Figure 2.

## 4. Discussion

### 4.1. Main Findings

Our data show associations of weekday eating habits with the prevalence of overweight in 4th- and 6th- to 8th-grade students in a city of about 600,000 inhabitants in Germany. Unsteady breakfast habits, skipping breakfast, or not having lunch during the school day were positively associated with the prevalence of being overweight. The associations with having breakfast persist after controlling for known risk factors like parental overweight and SES [28]. Having two breakfasts resulted in even stronger associations. The finding that children with a low SES skipped breakfast more often confirms our hypothesis as well as previous research [28,29,30,31,32,33,34,35].

### 4.2. Breakfast Behavior

More than two-thirds of the participating students reported regular breakfast consumption on school days, a rate similar to rates reported in other studies [28,36]. Moreover, our rate of about 6% complete breakfast skippers is similar to that of other studies from Europe and the US [28,37,38]. The observation that older students are more likely to skip breakfast than younger ones supports the study by Hallström, Monzani, Wadolowska and Smith [28,33,35,39]. Possible reasons might be the shrinking parental influence on adolescent eating behavior [36], the shortage of time due to late wake-up times [39,40], or dieting and weight-control behavior during adolescence [31]. Consistently, we found that children from secondary schools skipped breakfast more frequently than children from primary schools. This effect might be explained by the older age. In line with the systematic review by Monzani [39], we found that girls skipped breakfast more often than boys. Migration background was also associated with a higher risk of breakfast skipping, which is equivalent to the findings of Kesztyüs al. and Pedersen et al. [29,32].

Moreover, in another German study that found a significant association between breakfast skipping and weight status, the association lost statistical significance after adjusting for migration background, suggesting a potential mediating role of breakfast habits may be caused by cultural differences [41]. Of note, some children might not skip breakfast deliberately but are not offered breakfast because parents do not provide breakfast or do not consume breakfast themselves [14,36]. On the other hand, despite this being frequently reported [32,42], we could not confirm the association between parental overweight and breakfast skipping, a result which may be caused by a high proportion of missing data for parental overweight.

The proportion of participants who had two breakfasts exceeded the rates known from American literature [40,43]: while 47% of the students had one breakfast, another 47% had two breakfasts on their school days. A possible explanation is the overrepresentation of elementary school students in our population. For younger children, the reasons for breakfast skipping, as mentioned above, are less important.

According to a Swedish study, the energy not consumed during breakfast is not compensated by higher energy intake from lunch or dinner, but rather a higher intake of snack food and in-between meals [44]. Consistently, there have been associations shown between breakfast behavior and weight status, as discussed below.

### 4.3. Lunch Behavior

In our cohort, more than 80% of the students had regular lunch during school, which is in line with previous works [44,45,46]. We could confirm the results of Sjöberg et al., who found breakfast skipping to be associated with skipping other main meals [44], as we found that having breakfast increased the likelihood of having lunch. This strong link between breakfast and lunch habits may reflect an underlying trait of regular eating habits in general. As another work from Germany stated, breakfast intake alone does not explain the inverse association between higher meal frequency and childhood obesity [47]. Likewise, regular meal patterns might also be related to more general lifestyle habits [31,44]. The robust connection between having lunch and other health-related behavior could be one reason why the relation between having lunch and the risk of being overweight did not persist after controlling for other lifestyle factors, such as having breakfast.

### 4.4. Association of Breakfast and Lunch Behavior with Overweight

Our main results show that regular breakfast and lunch intake were associated with a lower BMI-SDS in children and adolescents. In line with Wang et al. and Bruening et al., we found students having two breakfasts had the lowest risk of being overweight, and breakfast skippers had the highest risk [40,43]. These findings could be explained due to the protective influence of a high meal frequency (e.g., five or more meals) on childhood obesity [47]. We speculate that a causal mechanism is the inverse association between meal frequency and mostly unhealthy snacking behavior. Skipping main meals, such as breakfast and lunch, was associated with a less healthy food choice, and the energy not consumed during breakfast was compensated by a rather higher intake from snack food instead of other main meals [44]. Given that the relation with having lunch could be explained by the covariates, the breakfast behavior seems to be of more importance. This is consistent with data from Gleason et al., who found an association of BMI with having school breakfast but not with having school lunch [48]. Moreover, Antonogeorgos et al. found that having regular meals during the day had only a positive health effect if children had breakfast [38].

We did not find any significant interaction, neither between the covariates and SES nor between the covariates and parental overweight. Contrarily, the multinational ISCOLE-study found a significant interaction between breakfast skipping and study site when modeling weight status [49]. In this work, the study sites represented a wide range of economic development and, hence, can be seen as an ecological proxy for SES. Beyond, although multiple studies reported that breakfast skipping is associated with both higher BMI and lower SES, in most of these studies, these associations were examined separately without reporting interactions [33,50].

In addition, having breakfast is associated with higher energy intake and the higher consumption of fruits, vegetables, and fiber [51]. As in our setting, measuring total energy uptake and the food composition was not feasible, we could not correct for these confounders. Moreover, we did not analyze dinner habits. However, our study is one of many providing evidence that having breakfast supports health-related behavior far beyond the mere energy intake facet [44,47,48].

It should be emphasized that the association between overweight and breakfast skipping could not be explained by the most common predictors of overweight, SES, and parental overweight [11,52,53]. Therefore, interventions targeting breakfast behavior provide a suitable starting point for weight gain interventions even more because breakfast can be integrated into the school day—and is therefore independent of the child´s social background. This approach is particularly promising given that children who had one breakfast on school days had on average a 0.5 lower BMI-SDS than children who had not. In comparison, conservative weight-loss treatments resulted in a weight loss in amounts from 0.05 to 0.39 one year after the start of the treatment in subjects that already suffered from overweight [54]. Thus, the effect size seems also clinically important. Subsidized meal programs that target breakfast participation in schools and childcare institutions are on the rise. While the federally assisted Breakfast Program in the US was permanently entitled since 1975 [55], the French Government has only recently introduced free breakfast in schools in prioritized areas [56]. Additionally, the US Department of Agriculture (USDA) specifies that schools and childcare institutions have to serve meals meeting federal nutrition requirements. Whether and to what extent a free school breakfast can lead to more favorable eating behavior and associated weight loss is not yet clearly established [48,57,58,59,60]. In contrast, a lack of governmental efforts in Germany may be seen. An EU school program supports schools in 12 of the 16 federal states to offer fruit, vegetables, and milk during the school day [61]. Together with the efforts of single non-profit organizations [62], this may not be to reach children and adolescents nationwide. Comprehensive, centrally funded solutions are desirable to avoid exacerbating the disadvantage of already deprived groups in the educational system.

### 4.5. Strength and Weaknesses

Our study examined a large sample size of elementary and secondary schoolchildren and their parents. However, the generalizability might be limited because the study population consisted largely of urban children. Moreover, the participants were of higher socio–economic status in comparison with the nationally representative sample [63]. However, this bias was smaller than in comparable studies due to the study design. Sampling within Leipzig was school centered rather than family centered, based on socio–economic data and the prevalence of overweight. Participation rates were even slightly higher than those of the KiGGS study, except for lower secondary school students [24,63]. This might be related to selective compliance, a common phenomenon in health survey participation causing the overrepresentation of the higher vs. the lower social strata [64,65]. Subsequently, the observed rate of overweight was slightly lower than the rates observed in a register-based study and the national reference study (~12% vs. ~17% and 15%) [2,4]. However, the rates are still at a comparable level. Moreover, assuming that non-participation was more likely among socially disadvantaged families, we would expect even stronger associations in the target population.

In addition, the school setting did not allow anthropometric measurements in underwear as recommended by the World Health Organization (WHO). However, the weight status of the children was assessed in a standardized weighing procedure rather than by self-reporting. The procedure included the adjustment for clothing. This method was used in previous research, and the inaccuracy of weight measures is reasonably small [18].

## 5. Conclusions

Our study reveals that a considerable proportion of children skipped breakfast, with differing rates depending on personal and family characteristics. Unsteady breakfast habits, skipping breakfast, or not having lunch were positively associated with overweight in schoolchildren. The associations with regular breakfast persisted regardless of parental overweight and SES, with having two breakfasts resulting in stronger associations. Therefore, interventions targeting healthier nutrition and its positive side effects by providing school breakfast seem to be promising, even more so because children of low SES are skipping breakfast more frequently. Such interventions can be implemented within the school setting and therefore, reach students across all social strata, independently of their home environments. Finally, our results confirm the school setting as a suitable starting point for community-based interventions and may underline the necessity of national programs providing free breakfast and lunch to children.

## Figures and Tables

**Figure 1 nutrients-13-01351-f001:**
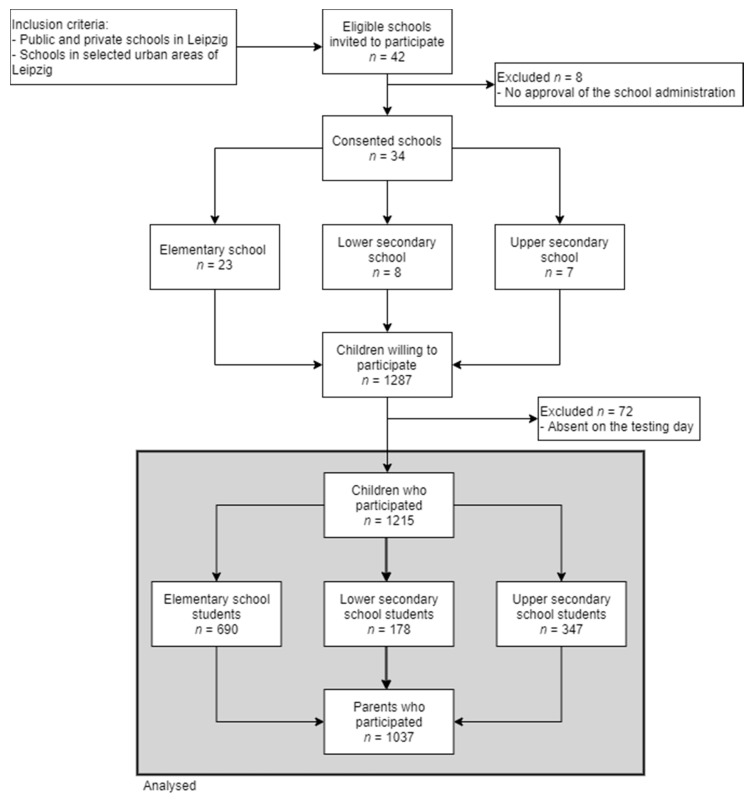
Flow diagram of inclusion and exclusion of study participants. Private schools combine different types of schools.

**Figure 2 nutrients-13-01351-f002:**
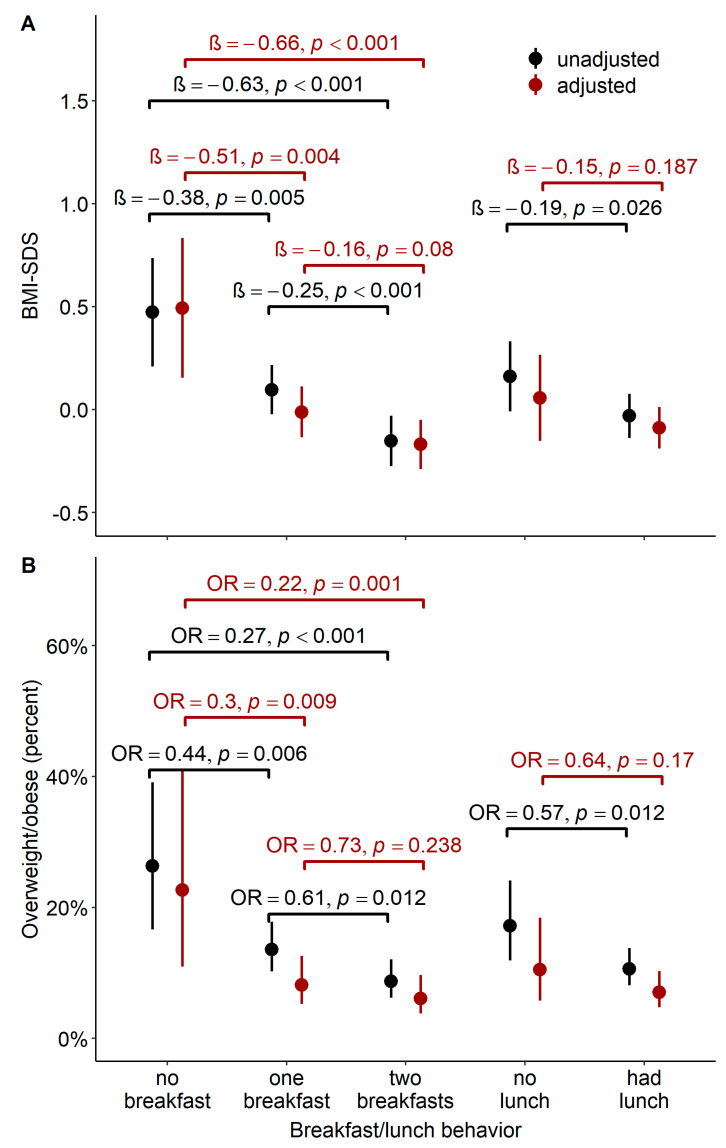
Association of different eating habits with mean BMI-SDS/the likelihood of being overweight. (**A**)**.** Mean BMI-SDS (incl. 95% CI) for children with different breakfast/lunch behavior: Based on the simple hierarchical regression analysis, children with different eating habits also differed in their mean BMI-SDS (black). Most of the associations persisted after adjustment (red). (**B**)**.** The likelihood of being overweight (incl. 95% CI) for children with different breakfast/lunch behavior: based on the simple hierarchical logistic regression analysis, children with different eating habits also differed in their likelihood of being overweight (black). The difference in the likelihood between children having and not having breakfast remained statistically significant after adjustment (red).

**Table 1 nutrients-13-01351-t001:** Description of the study sample.

Study Population (*n* = 1215)			
	*µ* (SD)	Range	*n* (%)
Age (years)	11.32 (1.35)	8.94–15.42	
Sex			
Male			598 (49.2)
Female			617 (50.8)
School type			
Elementary school			690 (56.8)
Lower secondary school			178 (14.7)
Upper secondary school			347 (28.6)
SES group			
Low			90 (10.7)
Medium			467 (55.3)
High			287 (34.0)
Missing			371
Migration background			
Yes			196 (19.6)
No			804 (80.4)
Missing			215
BMI-SDS	−0.04 (1.08)	−3.58–2.99	
BMI categorization			
BMI-SDS < 1.28			1063 (87.8)
BMI-SDS ≥ 1.28			148 (12.2)
Missing			4
Parental overweight/obesity (biological parents)			
Non			276 (30.9)
Both			180 (20.1)
Mother			155 (31.7)
Father			283 (17.3)
Missing			321

*µ*: mean; SD: standard deviation; *n*: count; BMI: body mass index; SES: socio–economic status.

**Table 2 nutrients-13-01351-t002:** Associations between regular breakfast skipping (*n* = 72) and different dependent variables. Effects are given as odds ratios (incl. 95% CI) for the unadjusted and the adjusted analyses.

Parameter		Breakfast Skipper (Ref: Breakfast Consumer)					
		Unadjusted OR			Adjusted OR		
		OR	95%-CI	*p*-Value	OR_adj_	95%-CI	*p*-Value
Age (per year)		1.54	(1.27–1.87)	<0.001			
Sex	Male	Ref.					
	Female	1.23	(1.23–1.24)	<0.001			
School type	Elementary school	Ref.					
	Secondary school	1.80	(1.11–2.91)	0.017	1.80 ^†^	(1.11–2.92)	0.016
SES group	Low	3.42	(1.56–7.46)	0.002	3.33	(1.55–7.14)	0.002
	Medium	Ref.					
	High	0.70	(0.31–1.60)	0.392	0.68	(0.30–1.54)	0.350
SES score		0.84	(0.77–0.93)	<0.001	0.85	(0.77–0.93)	<0.001
Migration background	No	Ref.					
	Yes	1.69	(0.87–3.26)	0.113	1.70	(1.69–1.71)	<0.001
Parental overweight (biological parents)	Not overweight/obese	Ref.					
	Both overweight/obese	0.95	(0.34–2.69)	0.928	0.99	(0.35–2.82)	0.986
	Mother overweight/obese	1.82	(0.71–4.67)	0.202	2.05	(0.80–5.29)	0.130
	Father overweight/obese	1.08	(0.44–2.66)	0.856	1.04	(0.43–2.56)	0.926

OR: odds ratio; OR_adj_: adjusted for age and sex; Ref: reference; †: adjusted only for sex.

## Data Availability

The legal requirements and the given informed consent do not allow public sharing of the dataset. Interested researchers can contact the research data management of the Medical Faculty, University Leipzig: forschungsdaten@medizin.uni-leipzig.de for further information. The dataset ID is PID-00080/01.

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
