# Peer review of "And yet Again: Having Breakfast Is Positively Associated with Lower BMI and Healthier General Eating Behavior in Schoolchildren"

_nutrients, 2021, doi:10.3390/nu13041351_

Round 1

Reviewer 1 Report

This is a very good manuscript on a hot topic. A well-planned test method. Correctly performed statistical analysis. The test results were presented clearly and reliably. A very good discussion. The only drawback is the correction of minor linguistic and stylistic errors, such as in table 1.

Author Response

Dear Reviewer,

We appreciate the time and effort you dedicated to providing feedback on our manuscript. We are grateful for the insightful comments and valuable improvements to our paper. We have incorporated most of the suggestions made by all reviewers. The changes are highlighted within the manuscript. Please see below for a point-by-point response. All line numbers refer to the revised manuscript file with tracked changes. We also provided a graphical abstract.

Best regards

Carolin Sobek, on behalf of all authors

Reviewer 1

This is a very good manuscript on a hot topic. A well-planned test method. Correctly performed statistical analysis. The test results were presented clearly and reliably. A very good discussion. The only drawback is the correction of minor linguistic and stylistic errors, such as in table 1.

  • Response: We are grateful for your comment and revised the entire manuscript regarding the stylistic and linguistic issues. In particular, in table 1: Column formatting; „Studypopulation“ to „study population“; font adapted; „Type of school“ to „school type“; „BMI-z-score“ to „BMI-SDS“. (table 1, line 166)

Reviewer 2 Report

This paper describes the relationship between breakfast, lunch and overweight in German school children. The manuscript is well developed, the objectives are clear, and the work well done. The results support the conclusions.

Table 1   Headline: Study population instead Studypopulation

Table 2   Title breakfast instead Breakfast

                Headline: Determinants of breakfast skipping instead                                                      Determinants of Breakfast skipping

Title

  • the title adequate to the content of publication

Materials and methods:

  • the methods are mostly appriopriate

Results:

  • the results are well discussed with the existing knowledge on the subject but no analyses for age, socio-economic status, school type, migration background and parental weight status in the discussion

Conclusions:

  • add data to show differences compared to the group that skipped breakfast,

 References:

  • the selection of literature and the presentation of literature data is appriopriate,

Author Response

Dear Reviewer,

We appreciate the time and effort you dedicated to providing feedback on our manuscript. We are grateful for the insightful comments and valuable improvements to our paper. We have incorporated most of the suggestions made by all reviewers. The changes are highlighted within the manuscript. Please see below for a point-by-point response. All line numbers refer to the revised manuscript file with tracked changes. We also provided a graphical abstract.

Best regards

Carolin Sobek, on behalf of all authors

  1. Results:

the results are well discussed with the existing knowledge on the subject but no analyses for age, socio-economic status, school type, migration background and parental weight status in the discussion

  • Response: Thank you for your valuable comment. Following your suggestions, we added the following information:

 Migration background: First, we clarified the usage of „migration background“ in the Methods. („Additionally, the migration background was asked for (“Yes” if at least one of the parents was born abroad).“ Page 3, line 121-2). We added one sentence in the Results („Girls skipped breakfast more often than boys (OR=1.23, 95%CI: 1.23-1.24, p<0.001), and migration background was also related to higher skipping rates (ORadj=1.70, 95%CI: 1.69-1.71, p<0.001)“ page 6, lines 180-2) and discussed the result in section 4.2; two citations were also added. The corresponding paragraphs now read as: „Migration background was also associated with a higher risk of breakfast skipping, which is equivalent to the findings of Kesztyüs al. and Pedersen et al. [29,32].

Moreover, in another German study that found a significant association between breakfast skipping and weight status, the association lost statistical significance after adjusting for migration background, suggesting a potential mediating role of breakfast habits may be caused by cultural differences [41].“ (page 9, lines 250-6)

 School type: We added the following sentence in Discussion 4.2: „Consistently, we found children from secondary schools skipping breakfast more frequently than children from primary schools. This effect is likely to be explained by the older age.“ (page 9, lines 247-49)

 Age: We discussed age in section 4.2. To point out the similarities between the results of previous studies and the current study, we changed the wording. The respective sentence was changed to: „The observation that older students are more likely to skip breakfast than younger students supports the study by Hallström, Monzani, Wadolowska, and Smith [28,33,35,39]“ (page 9, line 242-4)

 SES: We added the following sentence in the Discussion 4.1 Main Findings: „The finding that children with a low SES skip breakfast more often confirms our hypothesis as well as previous research [28-35].“ (page 9, lines 235-7)   

Besides, we added this sentence in the Conclusion: „Therefore, interventions targeting healthier nutrition and its positive side effects by providing school breakfast seem to be promising, more so because children of low-SES background are more often skipping breakfast.“ (page 12, lines 370-372)

 Parental weight status: We added the following sentence in the Results: „We did not find a statistically significant association between parental overweight and breakfast skipping, maybe due to a high number of missing parental overweight status (n=321).“ (page 6, lines 185-8) and discussed the result in 4.2: „On the other hand, despite frequently reported [32,42], we could not confirm the association between parental overweight and breakfast skipping, a result may be caused by a high proportion of missing data for parental overweight.“ (page 9, lines 259-62)

 SES and parental overweight as confounders of the association between breakfast skipping and weight status: We added the following sentences to the Discussion, section 4.4.

 „We did not find any significant interaction, neither between the covariates and SES nor between the covariates and parental overweight. Contrarily, the multinational ISCOLE-study found a significant interaction between breakfast skipping and study site when modeling weight status [49]. In this work, the study sites represented a wide range of economic development and, hence, can be seen as an ecological proxy for SES. Beyond, although multiple studies reported that breakfast skipping is associated with both lower BMI and lower SES, in most of these studies, these influences were examined separately without reporting interactions [33,50].“ (page 10, lines 301-8)

  1. Conclusions:

add data to show differences compared to the group that skipped breakfast,

  • Response: Thank you for this valuable comment. Indeed, a considerable proportion of the children skipped breakfast. We included the missing information in the Conclusion as follows:

Our study reveals that a considerable proportion of children skipped breakfast, with differing rates depending on personal and family characteristics.“ (page 11, lines 361-2)

  1. References:

the selection of literature and the presentation of literature data is appriopriate,

  • Response: Thank you for your kind comment.

Reviewer 3 Report

Thank you for the opportunity to review this manuscript. The topic is interesting; however, the authors may need clarify some issues.

My major concerns:

  1. Title

The title is “Yet again, breakfast is the most important meal of the day: The relationship between breakfast, lunch, and overweight in schoolchildren”, however, the authors focus on breakfast and lunch but not dinner. Is it appropriate to say breakfast is the most important meal of the day?

2.Methods

Cluster sampling was used to recruit participants in the study; thus, clustering effects may be adjusted for in the analysis.

  1. Methods

BMI-SDS was used as the main outcome and to define overweight/obesity. BMI Z-score would be a more relevant index to represent body composition, as BMI is sensitive to changes in age in children. The authors may need to analyse the association between breakfast/lunch habits and BMI Z-score, and overweight/obesity defined by BMI Z-score. The results can be put in the supplemental materials.

  1. Methods

Energy intake plays an important role in the development of obesity; however, total energy intake cannot be assessed in the study. The authors may need to discuss this.

  1. Results

Table 2 describes the “determinants” of regular breakfast skipping. The authors may consider toning down about this, given “determinants” were usually based on prospective studies or clinical trials.

  1. Discussion

Given the cross-sectional design of the study, the authors cannot say the “effect of lunch/breakfast” or something like that regarding their findings.

Minor concerns:

  1. Lines 150, 181, 184, 189: the authors mixed commas and minuses. They should check out these errors throughout the text.
  2. Figure 2: I am confused by the legends. The authors may need to modify the figure to clearly show the coefficients and odds ratios for the corresponding categories.
  3. “Effects” were frequently used to describe the associations in the Results. Effects of exposures were used to describe the intervention effects in clinical trials. The analysis was based on a cross-sectional study; thus, the authors may consider revising such statements.

Author Response

Dear Reviewer,

We appreciate the time and effort you dedicated to providing feedback on our manuscript. We are grateful for the insightful comments and valuable improvements to our paper. We have incorporated most of the suggestions made by all reviewers. The changes are highlighted within the manuscript. Please see below for a point-by-point response. All line numbers refer to the revised manuscript file with tracked changes. We also provided a graphical abstract.

Best regards

Carolin Sobek, on behalf of all authors

My major concerns:

  1. Title

The title is “Yet again, breakfast is the most important meal of the day: The relationship between breakfast, lunch, and overweight in schoolchildren”, however, the authors focus on breakfast and lunch but not dinner. Is it appropriate to say breakfast is the most important meal of the day?

  • Response: Thank you for pointing out this inaccuracy. We changed the title to:

„And yet again: Having breakfast is positively associated with lower BMI and healthier general eating behavior in schoolchildren“. (title, lines 4-6)

 Besides, we added the following sentence in the Discussion to clarify that we did not assess information about dinner behavior:

 „Moreover, we did not analyze dinner habits.“ (page 10, lines 312)

 Method

Cluster sampling was used to recruit participants in the study; thus, clustering effects may be adjusted for in the analysis.

  • Response: Thank you for this important comment. This certainly is an important point. We agree that schools are unique in the combination of circumstances that may are associated with attendance to school nutrition. We thus included school as a random effect in the models, changing the modeling from linear regression to hierarchical regression analyses. We added/changed the respective information in the Abstract, in the Methods, and adapted the Results section and the results in all Tables/Figures accordingly.

 See changes throughout paragraph 2.4 Statistical analyses.

“Models were adjusted for age and sex if necessary. To account for clustering effects, the school was added as a random effect. Due to the high collinearity leading to variance inflation, models containing school type as a covariate were not adjusted for age. Also, due to a strong dependence between school type and the school itself, the models with school type as the only covariate were modeled without random effect.” (page 3, lines 139-44)

 Methods

BMI-SDS was used as the main outcome and to define overweight/obesity. BMI Z-score would be a more relevant index to represent body composition, as BMI is sensitive to changes in age in children. The authors may need to analyse the association between breakfast/lunch habits and BMI Z-score, and overweight/obesity defined by BMI Z-score. The results can be put in the supplemental materials.

  • Response: Thank you for pointing this out. However, Z-scores and standard deviation scores (SDS) are based on the same statistical concept. Moreover, although Z-score is more frequently used if the assumed underlying distribution is the Normal distribution and standard deviation score in a context of more complex distributions, the definition is equivalent. We discussed the issue and decided to keep the wording „SDS“ because 1., the guidelines of the German Working Group of Obesity in Childhood and Adolescents (Wabitsch M. & Moss. A. 2019) uses this definition.

And 2., the underlying methods for creating the scores assumed a Box-Cox-Cole-and-Green distribution (Kromeyer-Hauschild, K et al. 2001; Cole T. 1990). They are also using „SDS“. However, to avoid confusion, we added some more information about SDS scores and added a sentence stating that SDS are equivalent to Z-scores in their interpretation.:

 „Subsequently, Body Mass Index (BMI) was calculated and transformed to standard deviation scores (SDS) using the references from Kromeyer-Hauschild [20], according to the German guidelines of the German Working Group of Obesity in Childhood and Adolescents [21]. SDS allow the assessment of weight status in the context of age and sex by transforming a BMI-value into its deviation from the age- and sex-dependent mean expectation, measured in multiples of a standard deviation. SDS are equivalent to Z-scores [22]. “ (page 2-3, lines 92-8)

  1. Methods

Energy intake plays an important role in the development of obesity; however, total energy intake cannot be assessed in the study. The authors may need to discuss this.

  • Response: Thank you for this comment. We absolutely agree that knowledge of pupils´ total energy uptake would have been an asset in this study. Thus, we added the following sentences in the Discussion:

“Besides, having breakfast is associated with higher energy intake and higher consumption of fruits, vegetables, and fiber [51]. Because in our setting, measuring total energy uptake and the food composition was not feasible, we could not correct for these confounders. Moreover, we did not analyze dinner habits. However, our study is one of many providing evidence that having breakfast supports health-related behavior far beyond the mere energy intake facet [44,47,48].” (page 10, lines 309-14)

  1. Results

Table 2 describes the “determinants” of regular breakfast skipping. The authors may consider toning down about this, given “determinants” were usually based on prospective studies or clinical trials.

  • Response: Thank you for this important comment! As suggested, we changed the wording.

„Determinants“ to „Parameter“ (page 6, line 178; table 2, line 198, heading and table header)

  1. Discussion

Given the cross-sectional design of the study, the authors cannot say the “effect of lunch/breakfast” or something like that regarding their findings.

  • Response: Thank you for this comment. You are absolutely right, and we changed the wording in the Abstract, in the Results, and adapted the Discussion and the Conclusion section accordingly. We now use the term „association with“ or „relation with“ throughout.

             Minor concerns:

  1. Lines 150, 181, 184, 189: the authors mixed commas and minuses. They should check out these errors throughout the text.
  • Response: Thank you. We reformatted the stylistic issue in line 150: „88% (n=607), 73.6% (n=131), 86.2% (n=299)“ to „88% (n=607)/ 73.6% (n=131)/ 86.2% (n=299)“ (page 4, line 159-60). In additon, we replaced the minuses in the Results section with hyphens „-“ to „–“.

 Figure 2: I am confused by the legends. The authors may need to modify the figure to clearly show the coefficients and odds ratios for the corresponding categories.

  • Response: Thank you for your comment. We agree that the figure legend did not describe the figure comprehensively enough. Therefore, we described the outcomes and comparisons depicted in a now separated legend for each subplot and hope to be clearer now.

 „Figure 2A. Mean BMI-SDS (incl. 95% CI) for children with different breakfast/lunch behavior: Based on the simple hierarchical regression analysis, children with different eating habits also differed in their mean BMI-SDS (black). Most of the associations persisted after adjustment (red).

Figure 2B. Likelihood of being overweight (incl. 95% CI) for children with different breakfast/lunch behavior: Based on the simple hierarchical logistic regression analysis, children with different eating habits also differed in their likelihood of being overweight (black). The difference in the likelihood between children having and not having breakfast remained statistically significant after adjustment (red).“  (table 2, lines 215-22)

  1. “Effects” were frequently used to describe the associations in the Results. Effects of exposures were used to describe the intervention effects in clinical trials. The analysis was based on a cross-sectional study; thus, the authors may consider revising such statements.
  • Response: Thank you! As suggested, we changed the wording throughout the manuscript.

Round 2

Reviewer 3 Report

I thank the authors for their responses to my comments. I had just one additional point, reviewing their revisions.

The authors may consider rewording the title of Table 2.